# De Novo Assembly, Characterization and Comparative Transcriptome Analysis of the Mature Gonads in *Spinibarbus hollandi*

**DOI:** 10.3390/ani13010166

**Published:** 2022-12-31

**Authors:** Chong Han, Wenwei Huang, Suhan Peng, Jiangwei Zhou, Huawei Zhan, Yuying Zhang, Wenjun Li, Jian Gong, Qiang Li

**Affiliations:** 1School of Life Sciences, Guangzhou University, Guangzhou 510006, China; 2Key Laboratory for Water Quality and Conservation of the Pearl River Delta, Ministry of Education, School of Environmental Science and Engineering, Guangzhou University, Guangzhou 510006, China

**Keywords:** *Spinibarbus hollandi*, transcriptome, gonad differentiation and development, sex

## Abstract

**Simple Summary:**

*Spinibarbus hollandi* has been recognized as an economically important aquaculture species in southeastern China. However, the information on its reproductive regulation is scarce. The long maturity period and low egg laying amount remain a major challenge for large-scale breeding of *S. hollandi*. In the current study, the gonad transcriptomes of *S. hollandi* were first obtained through Illumina Novaseq technology. By transcriptome comparing of ovary and testis, a lot of differential expression genes were identified and most of them were supposed to participate in gonad formation, differentiation and gametogenesis. Moreover, the expression levels of some typic reproductive genes indicated they might play similar roles in gonad differentiation and development of *S. hollandi*. Finally, the gonad transcriptome information can help researchers understand the regulatory functions of sex-related genes in *S. hollandi*.

**Abstract:**

*Spinibarbus hollandi* is an important commercial aquaculture species in southeastern China, but with long maturity period and low egg laying amount. However, there has been little study of its gonad development and reproductive regulation, which limits aquaculture production. Here, for the first time, gonadal transcriptomes of male and female *S. hollandi* were analyzed. A total of 167,152 unigenes were assembled, with only 48,275 annotated successfully. After comparison, a total of 21,903 differentially expressed genes were identified between male and female gonads, of which 16,395 were upregulated and 5508 were downregulated in the testis. In addition, a large number of differentially expressed genes participating in reproduction, gonad formation and differentiation, and gametogenesis were screened out and the differential expression profiles of partial genes were further validated using quantitative real-time PCR. These results will provide basic information for further research on gonad differentiation and development in *S. hollandi*.

## 1. Introduction

*Spinibarbus hollandi*, belonging to the subfamily Barbinae of Cyprinidae, is an important commercial aquaculture species in southeastern China, which is mainly distributed in Guangdong, Guangxi, Hunan, Hubei, Fujian, Anhui and Taiwan Provinces. *S. hollandi* has high nutritional value and medicinal value, and is popular as a table fish [1]. *S. hollandi* is an omnivorous species, which feeds mainly on aquatic insects, small fish, shrimp, and even blue-green algae and organic detritus [2]. Although *S. hollandi* is easy-to-raise, its growth rate is relatively slow, which limits the aquaculture efficiency and popularization of *S. hollandi* [3]. In recent years, due to its good taste and beautiful appearance, *S. hollandi* is increasingly becoming a popular fish with considerable economic value especially in south China. *S. hollandi* is a gonochoristic fish species, where males mature earlier than females. The age of first maturation of *S. hollandi* normally takes 3 to 4 years in south China. In addition, although *S. hollandi* is a cyprinid fish, only a few thousand eggs are laid at one time for female fish [4]. However, dozens to millions of eggs are laid at one time in grass carp and common carp. In short, long maturity period and low egg laying amount are critical factors limiting the yield increase of *S. hollandi*. Thus, it is extremely important to develop our understanding of the reproductive biology of this species.

For fish, gonadal development, differentiation and maturation are more complex than other vertebrates, which are easily influenced by the environmental factors [5]. In recent years, cumulative practices have shown that understanding the mechanisms of reproductive regulation is critically important for reproduction management and breeding practice. However, previous studies mainly focused on genetic resources, rearing conditions, morphology and molecular markers associated with growth of *S. hollandi* [3], whereas less attention was paid to gonadal development, differentiation, maturation and gametogenesis. Thus, to better understand the reproductive mechanisms and facilitate reproductive regulation, it is necessary to explore the functional genes associated with gonad development and differentiation, and gametogenesis.

Despite the rapid development of next generation sequencing (NGS) technology, the available genetic information on *S. hollandi* is rather limited. Only brain and muscle transcriptome data were published to explore the mechanism of starvation response and compensatory growth [6,7]. Up to now, nearly no reproduction-related gene was characterized and fundamental information of their expression is still lacking. Due to low cost and high throughput, NGS-based transcriptome sequencing has been a most effective method to generate a large number of transcripts and gene expression profiles in recent years [8]. It has been widely used for functional gene discovery and analysis of gene expression in teleosts. In addition, using gonad comparative transcriptome analysis, the expression patterns of reproduction-related genes have been revealed in many fish species such as Spot-fin porcupine fish (*Diodon hystrix*) [9], spotted scat (*Scatophagus argus*) [10], spotted knifejaw (*Oplegnathus punctatus*) [11], yellow catfish (*Pleteobagrus fulvidraco*) [12], mandarin fish (*Siniperca chuatsi*) [13] and so on. All these studies characterized a lot of candidate reproduction-related genes and provided basic information for reproductive regulation.

In this study, for the first time, Illumina-based transcriptome sequencing and de novo assembly of female and male gonads was carried out in *S. hollandi*. After comparative transcriptome analysis, a large number of sex-biased genes were uncovered and differentially expressed genes involved in reproductive regulation were also identified and discussed. Our data will enrich the currently available genetic and genomic data on *S. hollandi*, and identify candidate genes participating in gonad differentiation, development, maturation, and gametogenesis.

## 2. Materials and Methods

### 2.1. Sample Collection

In this study, six 4-year-old *S. hollandi* individuals (three males and three females) were obtained from an aquaculture farm in Shaoguan, Guangdong province. The fish were anesthetized with MS-222 before euthanizing them. Gonads from each individual were collected and stored in liquid nitrogen immediately. All animal handling procedures and experimental protocols were approved by the Experimental Animal Ethics Committee of the Guangzhou University of China (No. GURBBB210405).

### 2.2. RNA Extraction and Library Construction

Total RNA was extracted from fresh frozen tissue using RNA isolater Total RNA Extraction Reagent (Vazyme, Nanjing, China) according to the manufacturer’s instructions. The RNA concentration and purity were detected preliminarily using the Nanodrop2000 (Thermo Scientific, Wilmington, DE, USA). Then, integrity detection and quantification of RNA was carried out by Agilent 4200 Bioanalyzer (Agilent Technologies, Santa Clara, CA, USA). Eukaryotic transcriptome libraries were constructed using NEBNext^®^ Ultra™ RNA Library Prep Kit for Illumina^®^ (NEB, Ipswich, MA, USA). Eukaryotic mRNAs were enriched using magnetic beads with Oligo (dT). In a high temperature environment with metal ions, the RNA was fragmented and the first cDNA chain was synthesized with random hexamers, followed by the addition of enzyme, buffer, dNTPs (dATP, dTTP, dGTP, dCTP) to synthesis the second chain of cDNA. The synthesized double-stranded cDNA was purified by magnetic beads and repaired, and additional A was ligated to the tail end of cDNA, and the sequencing connector was connected. The sorted magnetic beads were used for fragment size sorting, and the sorted fragments were enriched by PCR, and finally purified to obtain the final library.

### 2.3. Library Sequencing, De Novo Assembly and Annotation

Using the Illumina Novaseq 6000 (Illumina, Inc., San Diego, CA, USA) high-throughput sequencing platform, the qualified libraries were sequenced with the sequencing strategy was Pair-End 150 (PE150). The amount of sequencing data for each library was not less than 6 Gb.

In order to obtain high quality assembly results for subsequent analysis, the raw reads were filtered, and low-quality reads with unknown base N beyond 5% and joint contamination were discarded. The remaining reads, also called “Clean reads”, were saved in FASTQ format. Then, using Trinity software, de novo assembly was carried out [14].

The coding region of Unigene was predicted by three forward and reverse reading frames, which could produce altogether 6 kinds of coding protein sequences. After obtaining the coding protein sequences, the protein sequences were compared with the non-redundant protein sequences (Nr) (https://www.ncbi.nlm.nih.gov/ (accessed on 20 April 2021)) and the Uniprot protein (https://www.uniprot.org/ (accessed on 20 June 2021)) database. The encoding method with the best alignment (the largest alignment score) was chosen as the encoding method of the gene.

Using homology searches, unigenes were annotated against five major public databases including Nr database, the Clusters of euKaryotic Orthologous Groups (KOG, http://www.ncbi.nlm.nih.gov/KOG (accessed on 27 June 2021)) database, the Kyoto Encyclopedia of Genes and Genomes (KEGG, http://www.genome.jp/kegg (accessed on 17 July 2021)) and the Uniprot protein database using BLASTx or Diamond. The NR results were further annotated by Blast2GO [15], and the unigenes were further grouped (GO, http://geneontology.org (accessed on 25 July 2021)) according to the classification of biological process, cell composition and molecular function.

### 2.4. Identification of Differentially Expressed Genes and Enrichment Analysis

The clean reads of each sample are first mapped to the assembled transcripts using hisat2 v2.1.0 [16]. Then, the transcripts per million (TPM), fragments per kilobase of exon model per million mapped fragments (FPKM), coverage and other gene-expression-related values were further calculated using stringtie v1.3.3b according to the comparison results [17]. Finally, using python prepDE.py, the output result of stringtie software was converted into a format recognized by the “edgeR” package (V3.6) [18,19], and the differential gene expression was analyzed using edgeR. The genes with *p*-value < 0.05 and |log2FC| > 2 are set as significantly differentially expressed genes (DEGs). The number of differentially expressed genes was counted by statistical significance, and then the clusterProfiler program of the R package [20] was applied against the background of DEGs. Based on Fisher’s exact test and Benjamini correction, the pathways that were significantly enriched in differentially expressed genes compared with the whole genome background with *p* value < 0.05 were considered to be significantly enriched.

### 2.5. Validation of DEGs Using Quantitative Real-Time PCR

The quantitative real-time PCR (qRT-PCR) analysis was used to validate the reliability of these DEGs. A total of 14 differentially expressed reproduction-related genes were randomly selected for validation. First, based on the sequences of 14 differentially expressed reproduction-related genes and reference gene β-actin, specific primers were designed by Primer Premier 5.0 (Table 1). Then, the HiScript III RT SuperMix for qPCR (+gDNA wiper) (Vazyme, Nanjing, China) was used for synthesizing cDNA template. Each qRT-PCR was carried out using SYBR Green qPCR Mix (GDSBIO, Guangzhou, China) on a LightCycle 480 system. The qRT-PCR reaction included an initial denaturation step at 95 °C for 3 min, followed by 40 cycles of 95 °C for 10 s, 60 °C for 20 s and 72 °C for 20 s, and a final extension at 72 °C for 5 min, ending with a dissociation curve process. Each sample was amplified in triplicate, and the expression of each gene was normalized using β-actin by the comparative CT method (2-ΔΔCT) [21] and was shown as mean ± standard error.

## 3. Results

### 3.1. Overview of Transcriptome Assembly Quality

A sum of six cDNA libraries was constructed in triplicates from testes (JC) and ovaries (LC). After quality control and data filtering, a total of 39.72 Gb clean reads was generated using an Illumina HiSeq platform, with a mean of 6.62 Gb, ranging from 6.0 to 7.28 Gb per sample. The mean values of Q20 and Q30 are higher than 95% (Table 2). After de novo assembly, 167,152 unigenes were assembled, with mean length and N50 of 871 bp and 1198 bp, respectively (Table 3). About 30.13% unigenes were in length of 300–500 bp, and 33,368 of them (19.97%) were more than 1000 bp in length (Figure 1).

### 3.2. Unigene Annotation

After annotation, a total of 48,275 unigenes were successfully annotated. The highest percentage of annotated unigenes was in the Uniprot database (47,343; 98.07%) and second is the Nr database (46,151; 95.60%), while the lowest was in the GO database (23,186; 48.03%) (Figure 2A). The distribution of BLASTx top-hit species showed that *Onychostoma macrolepis* (12,937; 28.03%) has the biggest number of homologous genes to *S. hollandi*, followed by *Cyprinus carpio* (5924; 12.84%), *Sinocyclocheilus rhinocerous* (5554; 12.03%) and *Sinocyclocheilus anshuiensis* (4699; 10.18%) (Figure 2B).

In addition, all unigenes were further annotated in GO, KOG and KEGG databases to get knowledge of their functional classification. A total of 23,186 (48.03%) unigenes were grouped into three categories of GO. In the biological processes category, cellular (19,188) was the most representative item. In the category of cellular component, the most represented term was cellular anatomical entity (22,121). In addition, in the molecular function category, the most representative terms were binding (17,708) and catalytic activity (11,582) (Figure 3A). After KEGG annotation, a sum of 25,817 (53.48%) unigenes was divided into five different functional categories. The largest distribution was “signal transduction” (3943 unigenes), “global and overview maps” (2552 unigenes) and “immune system” (2030 unigenes) (Figure 3B). According to KOG annotation, a total of 23,186 (48.03%) unigenes were grouped into 25 families, and the richest distribution was “General function prediction only” (5349 unigenes), followed by “Signal transduction mechanisms” (3452 unigenes); the smallest distribution was “Nuclear structure”, with only 8 unigenes (Figure 3C).

### 3.3. Differential Expression Analysis

The levels of gene expression were normalized using the TPM values. A total of 21,903 DEGs were identified between the ovary and testis samples, with 16,395 (74.85%) DEGs significantly high expressed in the testes and 5508 (25.15%) high expressed in the ovaries (Figure 4A). A volcano plot shows the DEG information (Figure 4B). The KEGG enrichment analysis revealed that the metabolic pathway was most representative, followed by the MAPK signaling pathway and the neuroactive ligand-receptor interaction pathway (Figure 5).

In addition, based on the GO and KEGG annotation, numerous genes related to reproduction and gonad development and differentiation were identified (Table 4). Anti-Mullerian hormone (*amh*), SRY-box transcription factor 9 (*sox9*), double sex- and mab-3-related transcription factor 2 (*dmrt2*) and androgen receptor (*ar*) were highly expressed in the testis, while cytochrome P450 (*cypa19a*), catenin beta-1 (*ctnnb1*) and bone morphogenetic protein 15 (*bmp15*) expressed highly in the ovary.

### 3.4. Validation of Transcriptomic Data

A total of fourteen DEGs related to sex differentiation and gonadal development were analyzed from the transcriptome data, including *amh*, *hsd11b2*, *fshr*, *era*, *erb1*, *erb2*, *ar*, *smad4a*, *gdf6*, *bmp7*, *pax7*, *pax8*, *sox5* and *sox9*. In general, the results of qRT-PCR and RNA-seq analysis were consistent (Figure 6), and the expression of all tested genes was higher in testes than in ovaries, which indicated the gene expression profiles quantified by transcriptomic analysis were reliable and accurate.

## 4. Discussion

Gonad development, differentiation and maturity are extremely complicated biological process, which contains a number of functional genes. In recent years, transcriptome sequencing has been demonstrated to be a highly effective method to obtain comprehensive information of gene expression for specific tissue. *S. hollandi* is an important commercial aquaculture species with high nutritional value and medicinal value. Its long sexual maturity time and low egg laying amount seriously hinder the development of the breeding industry. However, up to now, the molecular mechanisms of gonadal differentiation and development have not been revealed in *S. hollandi*. In this study, comparative transcriptome analysis of ovary and testis was first used to identify sex-related DEGs in *S. hollandi*.

### 4.1. DEGs Involved in Steroid Synthetic Pathway

Sex steroid hormones are critically important for gonadal differentiation, development and maturation in fish species, which mainly includes androgen and estrogen. In fish, the 11-ketotestosterone (11-KT) and 17-estradiol (E2) are the major androgen and estrogen that are essential to testicular and ovarian development [22]. In fact, sex steroid hormone synthesis needs a lot of steroid-metabolizing enzymes, which are encoded by many genes such as *cyp19a1a*, *cyp11b2*, *hsd11b1*, *cyp11a1*, *hsd17b* and so on.

Among them, the *cyp19a1a* gene is the most essential regulator, which can directly catalyze the switching of androgens to estrogens in the ovary. In *Oreochromis niloticus* and *Danio rerio*, knock out of the *cyp19a1a* gene caused a drop in E2 levels and further induced female-to-male sex reversal [23,24]. The *Hsd11b2* gene encodes an important enzyme that turns testosterone into 11-KT, which is important in testis maintenance. In *Danio rerio*, the reproductive capability of *hsd11b2* homozygous mutation adult males is almost completely abrogated [25]. The *Cyp11b1* gene is also involved in the synthesis of 11-KT, encoding the critical enzyme that turns testosterone to 11-KT [26]. In the sea bass, *cyp11b1* not only was predominantly expressed in the first stages of spermatogenesis but most likely was also expressed in spermatogonia [26]. In addition, high expression of *hsd11b2* and *cyp11b1* genes in the testis was also found in mandarin fish [27]. In *S. hollandi*, the expression of *cyp11b1* and *hsd11b2* genes was higher in the testis, while the *cyp19a1a* gene expressed highly in the ovary, indicating the expression pattern was similar among different fish species [9,10,27]. Thus, these results indicated that these DEGs play important roles in gonad development and reproduction in fish.

### 4.2. DEGs Involved in Gonad Differentiation and Development

In recent years, more and more genes were demonstrated to participate in the sex determination and gonad differentiation of teleosts. The upstream regulatory genes are diverse, but the downstream genes are much conserved. The common regulatory gene downstream of most sex-related genes is the *cyp19a1a* gene, which can directly catalyze the conversion of androgens to estrogens.

Anti-Mullerian hormone (*amh*), a transforming growth factor from the TGF-β family, controls the degeneration of female primordial germ cell ducts in mammals and promotes male development in vertebrates. Previous studies have demonstrated *amh* mutation could lead to male-to-female sex reversal in zebrafish [28] and tilapia [29]. Moreover, direct evidence revealed that *amh* could inhibit the transcription of *cyp19a1a* through *Amhr2/Smads* signaling [30]. *Bmp15* and *gdf9*, as oocyte-produced signals, participate in maintaining adult female sex differentiation and, in zebrafish, females deficient in *bmp15* begin development normally but switch sex during the mid- to late-juvenile stage, and become fertile males [31]. In *S. hollandi*, the results revealed that the expression of *amh* was higher in the testis, indicating the Amh pathway was of great importance in testicular differentiation regulation. In addition, *bmp15* and *gdf9* also showed ovary-biased expression in *S. hollandi*, suggesting a conserved role of *bmp15* and *gdf9* in teleosts.

Sox family genes are widely involved in the sex determination and differentiation process of vertebrates, and are a family of transcription factors that are closely related to the mammalian sex-determining region of the Y genes, namely *sry* [32]. Previous study has found that *sox5* could downregulate the activity of *dmy* (main sex determining gene) by binding to its promoter, and females with the *sox5* mutation will reverse to males in medaka [33]. *Sox9* is known to have an essential role in Sertoli cell differentiation; conditional *sox9* knockout testes fail to maintain germ cells in mice [34]. Previous evidence has proved that *sox9* is capable of interfering with WNT/β-catenin signaling and repressing the ovarian-differentiating genes including follistatin, Iroquois-related homeobox 3 and *foxl2* [35]. As in the medaka and mouse, *sox5* and *sox9* also showed high expression in the testis of *S. hollandi*, suggesting an important function of these genes in vertebrates.

### 4.3. DEGs Involved in Gametogenesis and Gamete Maturation

In aquaculture practice, it is extremely important to obtain parental fish with well-developed gametes. Many genes are involved in the regulation of gamete development and maturation. Here, the expression of many genes involved in gametogenesis and gamete maturation was also characterized, such as *spata16*, *zp3* and *zp4*. *Spata16*, namely spermatogenesis-associated protein 16, is essential for spermiogenesis [36] and male fertility; homozygous mutation of *spata16* leads to male infertility [37]. *Zp3* is a major kind of female-specific factor involved in the regulation of the reproductive process, forming an extracellular matrix surrounding oocytes which mediates sperm binding. Similar to *zp3*, *zp4* also surround oocytes and may act as a sperm receptor. In this study, the expression of *zp3* and *zp4* was higher in the ovary, indicating that they might participate in ovarian folliculogenesis in *S. hollandi*. In addition, *spata13*, *spag5* and *strbp* also showed higher expression in the ovaries, suggesting an important role in oogenesis and oocyte maturation. In all, similar expression patterns of these reproduction-related genes were also found in *Diodon hystrix* [9], *Scatophagus argus* [10], *Patinopecten yessonsis* [38] and so on, indicating a conserved role of these genes in teleosts.

## 5. Conclusions

This work is the first study on the gonad transcriptome of *S. hollandi*. Based on testis and ovary transcriptome data, a number of 167,152 unigenes were identified. Comparative transcriptome analysis identified a large scale of DEGs involved in gonadal development, differentiation and gametogenesis. For these DEGs related to gonad development and reproduction, similar expression profiles were found, suggesting their conserved roles in gonad development and gametogenesis. Our results will provide a valuable resource for further research on sex determination and gonad development.

## Figures and Tables

**Figure 1 animals-13-00166-f001:**
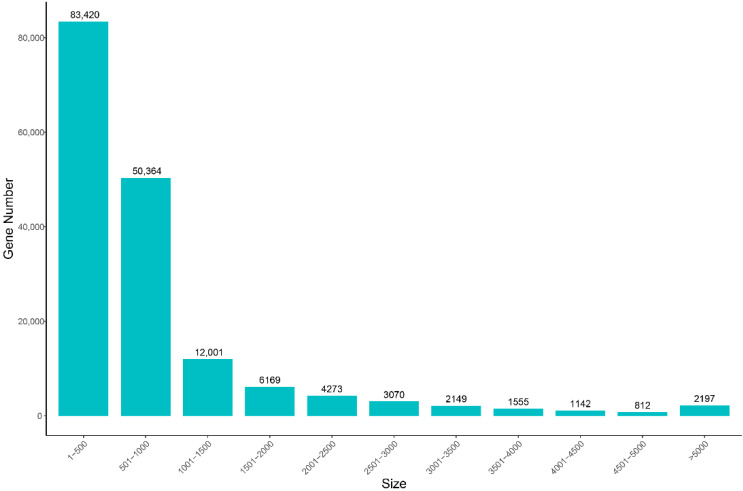
Length distribution of all assembled unigenes of the *S. hollandi* gonadal transcriptome. *X*-axis: size of unigene; *Y*-axis: number of unigenes.

**Figure 2 animals-13-00166-f002:**
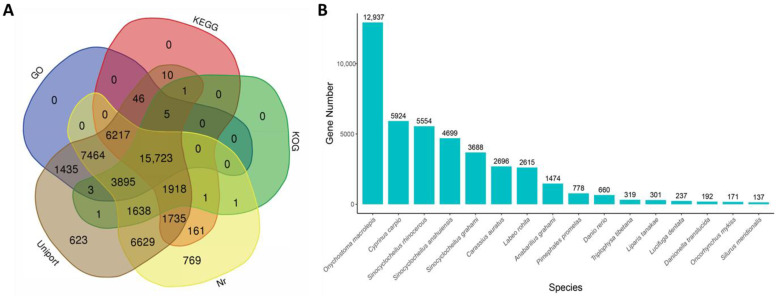
Venn diagrams of functional annotation against five major databases and species distribution of annotated homologous genes against NR database.

**Figure 3 animals-13-00166-f003:**
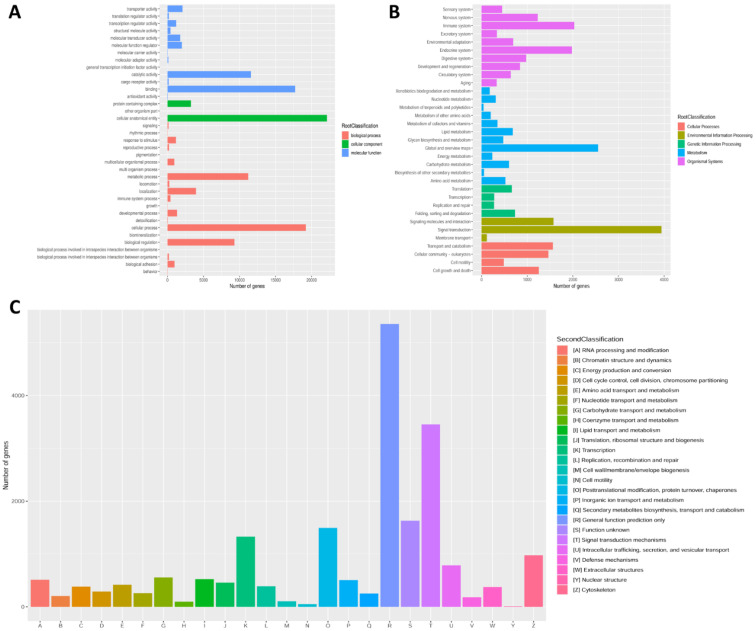
Gene function classification based on GO (**A**), KEGG (**B**) and KOG (**C**) databases.

**Figure 4 animals-13-00166-f004:**
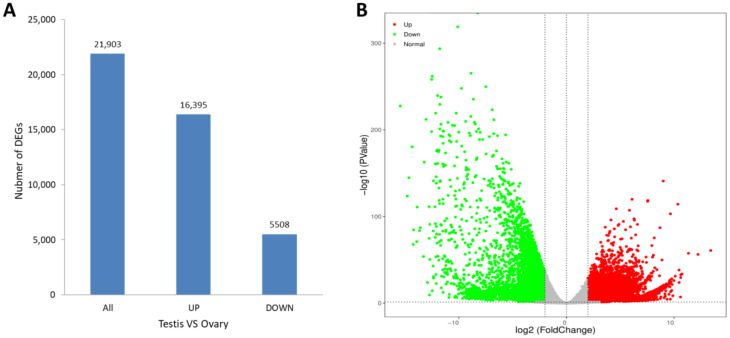
(**A**) Differentially expressed genes in males (testis) and females (ovary) of *S. hollandi*. (**B**) Volcano plot of DEGs in testis versus ovary. Red and green keys and dots represent upregulated and downregulated genes, respectively, in males, and vice versa.

**Figure 5 animals-13-00166-f005:**
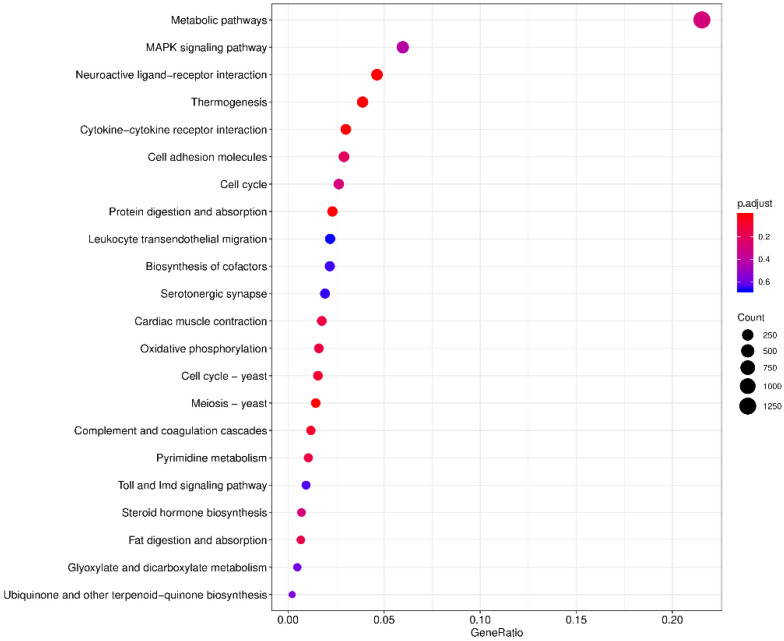
Top 22 KEGG enrichment analysis. The horizontal axis is the ratio of the number of differential genes annotated to the KEGG pathway to the total number of differential genes. The vertical is the enriched KEGG pathways. The size of the dots represents the number of genes annotated on the KEGG pathway term. The colors key, from red to blue, represent significant enrichment.

**Figure 6 animals-13-00166-f006:**
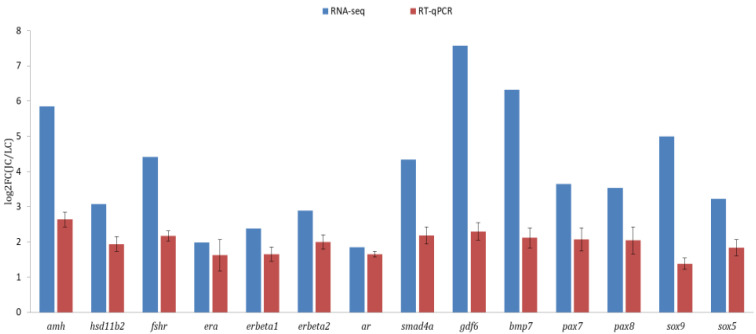
Validation of gonad transcriptome data by qRT-PCR.

**Table 1 animals-13-00166-t001:** Primers of qRT-PCR.

Primer ID	Primer Sequences
GDCB-amh-QP1F	TCCACAGAGTCCCTGCTATCA
GDCB-amh-QP1R	CAATCACCTTTTGCCCCAC
GDCB-hsd11b2-QP1F	AGACAGGCTAAAGGCCGGA
GDCB-hsd11b2-QP1R	TGACGAAGTGTGTTGGTAAGAAGAT
GDCB-fshr-QP1F	CGCCTTCTTCAACCTCAC
GDCB-fshr-QP1R	GTTACTTGGAATCCTTTCTATG
GDCB-era-QP1F	GCATTTTCATTCTGCTCCAGTC
GDCB-era-QP1R	CATTCCTTTGTTGCTCATGTGTCT
GDCB-erbeta1-QP1F	CCGCTGGTGAAGAGGGTAGT
GDCB-erbeta1-QP1R	GCAGATGTAGTCATTGTGTCCTTGTAT
GDCB-erbeta2-QP1F	TGGAGAGCCGTGGGAAGGTT
GDCB-erbeta2-QP1R	TGTTGCTGAGGTGTCGAATATG
GDCB-ar-QP1F	CCCCCAGCAAGGGAATAATATG
GDCB-ar-QP1R	TCAGCGCCCCGAAAGAAA
GDCB-foxk2-QP1F	TGAAGCTGCCGAGAGTGTGC
GDCB-foxk2-QP1R	GGGATGTTGATGGTTAAAGGGG
GDCB-smad4a-QP1F	CCAGGGAGGAGAGAATGAGAGC
GDCB-smad4a-QP1R	CGAGCGTATATGACATGAGGGAA
GDCB-gdf6-QP1F	AGTGTCCAAGTCTGTTCCTGCT
GDCB-gdf6-QP1R	CTCGTTATGGTGTTTGCTGC
GDCB-bmp7-QP1F	CACCGCTCTAAAAACCCCA
GDCB-bmp7-QP1R	ATTCTCCCTCACAGTAATACGCA
GDCB-pax7-QP1F	AGGCCAGAGTGCAGGTTTGG
GDCB-pax7-QP1R	TGTGGTAATGTTGTTGAAGGATAGG
GDCB-pax8-QP1F	AAACCAGGAGTCATCGGGGG
GDCB-pax8-QP1R	CCTTTGCTGTCCAGAGGAAGG
GDCB-sox9-QP1F	AGGTCAGAGCTCCGGCTTGTACT
GDCB-sox9-QP1R	TGTGATTGGGTTGGGGAATGG
GDCB-sox5-QP1F	ACCAAAGCCCAAGGACGAGG
GDCB-sox5-QP1R	GCCGGAGGTGATGGATGACA
GDCB-beta-actin-QP1F	GTGTTGGCATACAGGTCCTTACG
GDCB-beta-actin-QP1R	ACGGACAGGTCATCACCATTG

**Table 2 animals-13-00166-t002:** Summary statistics of the gonadal RNA-Seq data.

Sample	Reads Number	Base Number	Total Base	Q20	Q30	GC
JC-K5	23,015,034	3,452,255,100	6,904,510,200	98.33%	95.25%	46.53%
	23,015,034	3,452,255,100		97.96%	94.12%	46.51%
JC-K8	24,277,601	3,641,640,150	7,283,280,300	98.29%	95.16%	46.56%
	24,277,601	3,641,640,150		97.86%	93.88%	46.54%
JC-K9	19,990,173	2,998,525,950	5,997,051,900	98.32%	95.21%	46.21%
	19,990,173	2,998,525,950		98.08%	94.42%	46.18%
LC-K1	23,109,316	3,466,397,400	6,932,794,800	98.11%	94.96%	48.05%
	23,109,316	3,466,397,400		97.85%	94.20%	48.03%
LC-K3	21,181,082	3,177,162,300	6,354,324,600	98.29%	95.23%	48.17%
	21,181,082	3,177,162,300		97.83%	94.01%	48.15%
LC-K9	20,827,911	3,124,186,650	6,248,373,300	98.42%	95.45%	48.03%
	20,827,911	3,124,186,650		98.12%	94.58%	48.00%

**Table 3 animals-13-00166-t003:** Summary statistics of the *S. hollandi* gonadal transcriptome assembly.

	Number
Gene Number (#):	167,152
Total length (nt):	145,709,933
Average length (nt):	871
Max length (nt):	18,177
Min length (nt):	280
N50	1198

**Table 4 animals-13-00166-t004:** The patterns of differentially expressed genes related to reproduction, gonad development and differentiation in the gonads of *S. hollandi* (JC_vs_LC means testis vs ovary).

Unigene ID	log2FC (JC_vs_LC)	*p* Value (JC_vs_LC)	FDR (JC_vs_LC)	Nr Annotation	Gene Name
unigene074455	7.572	4.92 × 10^−9^	1.83 × 10^−08^	Growth differentiation factor 6	*gdf6*
unigene027408	7.318	2.71 × 10^−9^	1.04 × 10^−08^	Neuropeptide Y receptor type 2	*npy2r*
unigene016644	7.159	7.36 × 10^−14^	4.50 × 10^−13^	Cytochrome P450 11B	*cyp11b1*
unigene058283	6.803	4.63 × 10^−08^	1.52 × 10^−07^	Forkhead box protein K2	*foxk2*
unigene065602	6.718	3.09 × 10^−04^	5.54 × 10^−04^	Paired box protein Pax-6	*pax6*
unigene047253	6.481	4.79 × 10^−06^	1.16 × 10^−05^	Fibroblast growth factor 20	*fgf20*
unigene082603	6.325	2.40 × 10^−05^	5.23 × 10^−05^	Bone morphogenetic protein 7	*Bmp7*
unigene100115	6.112	1.68 × 10^−04^	3.17 × 10^−04^	Forkhead box G1-like protein	*foxg1*
unigene034505	6.052	1.43 × 10^−05^	3.24 × 10^−05^	Catenin delta-2	*ctnnd2*
unigene022426	5.941	1.44 × 10^−19^	1.38 × 10^−18^	Paired box protein Pax-2a	*pax2*
unigene141021	5.878	2.57 × 10^−12^	1.37 × 10^−11^	Nanos homolog 2	*nanos2*
unigene021432	5.874	5.49 × 10^−05^	1.12 × 10^−04^	Paired box protein Pax-7	*pax7*
unigene008910	5.867	3.93 × 10^−05^	8.24 × 10^−05^	Forkhead box protein L1	*foxl1*
unigene106729	5.856	5.75 × 10^−108^	1.08 × 10^−105^	Anti-Mullerian hormone	*amh*
unigene035421	5.798	7.51 × 10^−13^	4.21 × 10^−12^	Insulin-like growth factor-I	*igf1*
unigene086594	5.661	1.96 × 10^−10^	8.53 × 10^−10^	Protein Wnt-6	*wnt6*
unigene130369	5.617	2.78 × 10^−03^	4.27 × 10^−03^	Transforming growth factor beta receptor type 3	*tgfbr3*
unigene123135	4.990	7.83 × 10^−33^	1.54 × 10^−31^	Transcription factor SOX9	*sox9*
unigene089170	4.878	1.56 × 10^−13^	9.30 × 10^−13^	Protein Wnt-5a	*wnt5*
unigene025970	4.539	6.90 × 10^−11^	3.16 × 10^−10^	Protein Wnt-2b	*wnt2*
unigene009697	4.487	1.83 × 10^−10^	8.00 × 10^−10^	Gonadotropin-releasing hormone II receptor	*gnrhr*
unigene125833	4.461	1.52 × 10^−19^	1.45 × 10^−18^	Sperm-associated antigen 1	*spag1*
unigene159834	4.418	1.43 × 10^−06^	3.77 × 10^−06^	Follicle stimulating hormone receptor	*fshr*
unigene018630	4.344	6.47 × 10^−23^	7.59 × 10^−22^	Mothers against decapentaplegic homolog 4a	*smad4a*
unigene143320	4.016	5.73 × 10^−06^	1.38 × 10^−05^	Forkhead box D2-like protein	*foxd2*
unigene036991	3.766	1.74 × 10^−04^	3.27 × 10^−04^	Protein Wnt-3a	*wnt3*
unigene131983	3.725	1.54 × 10^−09^	6.08 × 10^−09^	Bone morphogenetic protein 6 isoform X1	*bmp6*
unigene072066	3.692	1.06 × 10^−04^	2.07 × 10^−04^	Forkhead box P1-B-like isoform X1	*foxp1*
unigene021595	3.626	2.58 × 10^−06^	6.54 × 10^−06^	Bone morphogenetic protein receptor type-2	*bmpr2*
unigene094351	3.575	1.65 × 10^−09^	6.47 × 10^−09^	Paired box protein Pax-3	*pax3*
unigene077500	3.532	5.58 × 10^−08^	1.81 × 10^−07^	Paired box protein Pax-8	*pax8*
unigene114563	3.461	6.23 × 10^−06^	1.49 × 10^−05^	Forkhead box protein O6	*foxo6*
unigene020954	3.447	2.05 × 10^−07^	6.11 × 10^−07^	Transcription factor Sox-1a	*sox1*
unigene151019	3.352	6.41 × 10^−08^	2.06 × 10^−07^	Protein Wnt-11	*wnt11*
unigene157484	3.304	7.84 × 10^−33^	1.54 × 10^−31^	Fibroblast growth factor receptor 2	*fgfr2*
unigene060599	3.220	2.39 × 10^−18^	2.10 × 10^−17^	Transcription factor SOX-5	*sox5*
unigene060624	3.212	9.24 × 10^−11^	4.17 × 10^−10^	Growth-hormone-releasing hormone receptor	*ghrhr*
unigene071307	3.210	6.50 × 10^−17^	5.12 × 10^−16^	Transforming growth factor beta-2	*tgfb2*
unigene119381	3.181	1.92 × 10^−10^	8.37 × 10^−10^	Insulin-like growth-factor-binding 3	*igfb3*
unigene002746	3.074	6.17 × 10^−20^	6.04 × 10^−19^	Corticosteroid 11-beta-dehydrogenase isozyme 2	*hsd11b2*
unigene073844	2.890	4.90 × 10^−13^	2.79 × 10^−12^	Estrogen receptor beta 2	*erb2*
unigene076208	2.764	1.34 × 10^−06^	3.55 × 10^−06^	Follitropin subunit beta	*fshb*
unigene145623	2.763	7.92 × 10^−09^	2.87 × 10^−08^	Insulin-like growth-factor-binding protein 1	*igfbp1*
unigene144860	2.759	2.51 × 10^−07^	7.39 × 10^−07^	Forkhead box protein O3	*foxo3*
unigene052054	2.730	6.22 × 10^−11^	2.86 × 10^−10^	Fibroblast growth factor receptor 4	*fgfr4*
unigene126929	2.604	6.48 × 10^−06^	1.55 × 10^−05^	Bone morphogenetic protein 2	*bmp2*
unigene146529	2.383	7.29 × 10^−07^	2.01 × 10^−06^	Transcription factor sox7	*sox7*
unigene156535	2.380	2.36 × 10^−13^	1.38 × 10^−12^	Estrogen receptor beta 1	*erb1*
unigene057687	2.372	8.63 × 10^−07^	2.35 × 10^−06^	stAR-related lipid transfer protein 13	*stard13*
unigene031051	2.351	1.86 × 10^−05^	4.13 × 10^−05^	Forkhead box protein D1	*foxd1*
unigene013811	2.265	3.43 × 10^−03^	5.18 × 10^−03^	Androgen receptor protein	*ar*
unigene101954	2.264	2.05 × 10^−04^	3.80 × 10^−04^	Gonadotropin-releasing hormone	*gnrh*
unigene100104	2.236	3.13 × 10^−06^	7.84 × 10^−06^	double sex- and mab-3-related transcription factor 2	*dmrt2*
unigene008405	2.207	1.52 × 10^−03^	2.43 × 10^−03^	Spermatogenesis-associated protein 16	*sgap16*
unigene105851	2.174	3.57 × 10^−03^	5.37 × 10^−03^	Paired box protein Pax-4	*pax4*
unigene040229	2.160	1.35 × 10^−08^	4.77 × 10^−08^	Growth/differentiation factor 10	*gdf10*
unigene122511	2.117	1.60 × 10^−06^	4.18 × 10^−06^	Transcription factor SOX-4	*sox4*
unigene054659	−2.416	3.36 × 10^−13^	1.94 × 10^−12^	Spermatid perinuclear RNA-binding protein isoform X3	*strbp*
unigene147986	−2.463	3.91 × 10^−09^	1.47 × 10^−08^	Cytochrome P450 aromatase	*cyp19a1a*
unigene017060	−2.563	1.13 × 10^−31^	2.10 × 10^−30^	TGF-beta receptor type-2-like isoform X1	*tgfbr2*
unigene124450	−2.947	2.27 × 10^−38^	5.69 × 10^−37^	Catenin beta-1	*ctnnb1*
unigene062930	−2.964	5.89 × 10^−11^	2.72 × 10^−10^	Transcription factor Sox-3	*sox3*
unigene014765	−3.027	4.13 × 10^−10^	1.73 × 10^−09^	Spermatogenesis-associated protein 13	*sgap13*
unigene000100	−3.070	8.27 × 10^−07^	2.26 × 10^−06^	Sperm-associated antigen 5	*spag5*
unigene127688	−3.105	3.71 × 10^−44^	1.16 × 10^−42^	sTART domain-containing protein 10	*stard10*
unigene152354	−3.674	8.97 × 10^−10^	3.63 × 10^−09^	Paired box protein Pax-1	*pax1*
unigene051443	−3.717	2.71 × 10^−18^	2.37 × 10^−17^	Double sex- and mab-3-related transcription factor A1	*dmrta1*
unigene143561	−3.915	1.79 × 10^−71^	1.39 × 10^−69^	Growth/differentiation factor 9	*gdf9*
unigene125277	−3.925	5.06 × 10^−13^	2.88 × 10^−12^	Estradiol 17-beta-dehydrogenase 2	*hsd17b2*
unigene069571	−3.965	1.60 × 10^−88^	1.96 × 10^−86^	Bone morphogenetic protein 15	*bmp15*
unigene086306	−4.644	2.71 × 10^−18^	2.37 × 10^−17^	Nuclear receptor subfamily 5 group A member 2	*nr5a2*
unigene007641	−5.956	5.52 × 10^−41^	1.55 × 10^−39^	Forkhead box protein P3	*foxp3*
unigene060440	−5.993	3.07 × 10^−50^	1.20 × 10^−48^	Transcription factor Sox-19b	*sox19b*
unigene150909	−6.870	4.59 × 10^−170^	3.12 × 10^−167^	Transcription factor Sox-21-A	*sox31a*
unigene158966	−7.139	8.15 × 10^−47^	2.82 × 10^−45^	Protein Wnt-8a	*wnt8*
unigene043515	−11.722	3.97 × 10^−142^	1.52 × 10^−139^	Zona pellucida sperm-binding protein 4	*zp4*
unigene055600	−12.153	5.03 × 10^−220^	1.32 × 10^−216^	Zona pellucida sperm-binding protein 3	*zp3*
unigene077416	−14.328	3.82 × 10^−181^	3.10 × 10^−178^	Growth/differentiation factor 3	*gdf3*

## Data Availability

The raw data are available from the SRA (http://www.ncbi.nlm.nih.gov/sra/, accessed on 12 October 2021) data repository (accession number: PRJNA907069).

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
