# Peer review of "De Novo Assembly, Characterization and Comparative Transcriptome Analysis of the Mature Gonads in Spinibarbus hollandi"

_animals, 2022, doi:10.3390/ani13010166_

Round 1

Reviewer 1 Report

Spinibarbus hollandi, is an important commercial aquaculture species in southeastern China. However, there has been less related studies. This work is the first study on the gonad transcriptome of S. hollandi. Comparative transcriptome analysis identified a large scale of DEGs involved in gonadal development, differentiation and gametogenesis. For these DEGs related to gonad development and reproduction, similar expression profiles were found, suggesting their conserved roles in gonad development and gametogenesis. These results would provide a valuable resource for further research on the sex determination and gonad development.

Question: why estrogen related receptor such as era1, erbeta1 and erbeta2 expressed highly in testis?

some minor errors as follow:

Line 177: it should be “Reads Number, Base number”

Line 242: “differentially expressed reproduction and gonad development and differentiation related genes” should be “differentially expressed genes related to reproduction, gonad development and differentiation”

Line 291: “could” should be “can”.

Author Response

Dear Reviewer:

Thank you for your letter and for the reviewers’ comments concerning our manuscript entitled “De Novo Assembly, Characterization and Comparative Transcriptome Analysis of the Mature Gonads In Spinibarbus hollandi” (ID: animals-2103060). Those comments are all helpful for revising and improving our paper, as well as the important guiding significance to our researches. We have studied comments carefully and have made correction which we hope meet with approval. Revised portion are marked in red in the paper. The main corrections in the paper and the responds to the reviewer’s comments are as following:

List of Responses:

Response:

Comment 1. Question: why estrogen related receptor such as era1, erbeta1 and erbeta2 expressed highly in testis?

Response: Thank you for your question. In fact, the high expression of era and erbeta has been widely reported in several species, such as mandarin fish (Siniperca chuatsi) (Ouyang et al., 2021), European sea bass (Dicentrarchus labrax) (Blazquez et al., 2008). Even in tilapia, erβ1 knockout male fish showed fewer spermatogonocytes and more abnormal sperm, and erβ2 knockout male fish showed malformation of reproductive tract, further suggesting that erβ1 and erβ2 play an important role in spermatogenesis and male reproductive organ formation (Yan et al., 2019).

Ouyang, H., Han, C., Zhu, Q., Xu, L., Huang, J., Li, S., et al. (2021). Molecular cloning and characterization of estrogen and androgen receptors in Mandarin fish, Siniperca chuatsi. Aquaculture Reports, 21, 100834.

Blazquez, M., Gonzalez, A., Papadaki, M., Mylonas, C., Piferrer, F., 2008. Sex-related changes in estrogen receptors and aromatase gene expression and enzymatic activity during early development and sex differentiation in the European sea bass (Dicentrarchus labrax). Gen. Comp. Endocrinol. 158, 95-101.

Yan L, Feng H, Wang F, et al. Establishment of three estrogen receptors (esr1, esr2a, esr2b) knockout lines for functional study in Nile tilapia. The Journal of Steroid Biochemistry and Molecular Biology, 2019, 191: 105379.

Comment 2. some minor errors as follow:

Line 177: it should be “Reads Number, Base number”

Line 242: “differentially expressed reproduction and gonad development and differentiation related genes” should be “differentially expressed genes related to reproduction, gonad development and differentiation”

Line 291: “could” should be “can”.

Response: Thank you for your correction. We have modified these errors.

We tried our best to improve the manuscript and made some changes in the manuscript. These changes will not influence the content and framework of the paper. We appreciate for Editors/Reviewers’ warm work earnestly, and hope that the correction will meet with approval.

Once again, thank you very much for your comments and suggestions.

Yours

Sincerely

Qiang Li

Reviewer 2 Report

This study aims to form a basis for developing our understanding of the reproductive biology of a commercially important fish, Spinibarbus hollandi, by exploring its transcriptome. The authors generate the transcriptome and validate through RT-qPCR. The work is fairly comprehensive, with a robust method, and a link to appropriate ethical approval provided. The genes identified are placed into context within the Discussion. Most of my comments focus on formatting and presentation, but I feel this would be a useful addition to the literature.

Introduction

• L51: Change "In a word" to "In short"

• L52: Change to "Thus, it's extremely important to develop our understanding of reproductive biology of this species." 

• The term "reproduction regulation" might need to be changed to "reproductive regulation".

Methods

• L. 118-120: I think there are some conjoined sentences here, as this doesn't make much sence. 

• Table 1 is fine, really, but may read a little more easily if primer pairs are more clearly identified e.g. a small gap between pairs, alternating white and grey backrground for each pair, presenting the table in wide format with each pair on the same row, or something like that. But this is just a suggestion for clarity.

• Table 3: Either have the data here or in the text, there is no need to repeat this information.

Results

• Figure 3: The text size is necessarily small... But the classification legends in particular are very difficult to read, perhaps due to the font. 

• Table 4: Only a suggestion but I would find small numbers presented in standard format easier to read than in E format. In the title indiciate that genes are presented in order of fold change. Indicate what JC and LC mean.

• Figure 6: There are quite large differences in the differences in expression (fold change) depending on method. For example, fold change was equivalent between RNA-seq and RT-qPCR for era and ar, but there is a fc difference of up to 5 or 6 in some other genes such as as gdf6 and bmp7.Fold change in RNA-seq was much more variable than in RT-qPCR, and in some cases the order of expression differs depending on method (e.g. sox9 has one of the lowest fold change values in RT-qPCR but one of the highest for RNA-seq). How do the authors define consistency here? This might all be fine but the process needs to be clear to readers.

Discussion

• L. 273-275: could this have a reference?

Author Response

Dear Reviewer:

Thank you for your letter and for the reviewers’ comments concerning our manuscript entitled “De Novo Assembly, Characterization and Comparative Transcriptome Analysis of the Mature Gonads In Spinibarbus hollandi” (ID: animals-2103060). Those comments are all helpful for revising and improving our paper, as well as the important guiding significance to our researches. We have studied comments carefully and have made correction which we hope meet with approval. Revised portion are marked in red in the paper. The main corrections in the paper and the responds to the reviewer’s comments are as following:

List of Responses:

Response:

Comment 1. Introduction

  • L51: Change "In a word" to "In short"

  • L52: Change to "Thus, it's extremely important to develop our understanding of reproductive biology of this species."
  • The term "reproduction regulation" might need to be changed to "reproductive regulation".

Response: Thank you for your correction. We have modified these errors.

Comment 2. Methods

  • L. 118-120: I think there are some conjoined sentences here, as this doesn't make much sence.
  • Table 1 is fine, really, but may read a little more easily if primer pairs are more clearly identified e.g. a small gap between pairs, alternating white and grey backrground for each pair, presenting the table in wide format with each pair on the same row, or something like that. But this is just a suggestion for clarity.
  • Table 3: Either have the data here or in the text, there is no need to repeat this information.

Response: Thank you for your correction. We have deleted some conjoined sentences in L.118-120. For table1, we have enlarged the table a little. For table 3, it seems that it is necessary to show it in a table.

Comment 3. Results

  • Figure 3: The text size is necessarily small... But the classification legends in particular are very difficult to read, perhaps due to the font.
  • Table 4: Only a suggestion but I would find small numbers presented in standard format easier to read than in E format. In the title indiciate that genes are presented in order of fold change. Indicate what JC and LC mean.
  • Figure 6: There are quite large differences in the differences in expression (fold change) depending on method. For example, fold change was equivalent between RNA-seq and RT-qPCR for era and ar, but there is a fc difference of up to 5 or 6 in some other genes such as as gdf6 and bmp7. Fold change in RNA-seq was much more variable than in RT-qPCR, and in some cases the order of expression differs depending on method (e.g. sox9 has one of the lowest fold change values in RT-qPCR but one of the highest for RNA-seq). How do the authors define consistency here? This might all be fine but the process needs to be clear to readers.

Response: Thank you for your suggestion. For figure 3, we have replaced it by a clear one. For table 4, we have added the information on what JC and LC mean. For figure 6, the consistency mainly means the expression similarity of trends between testis and ovary. It is difficult to guarantee the the complete consistency of the results based on different methods. Besides, in fact, for most genes, the expression profiles were similar based on different methods in our study. We need to respect the similarity and difference of different methods.

We tried our best to improve the manuscript and made some changes in the manuscript. These changes will not influence the content and framework of the paper. We appreciate for Editors/Reviewers’ warm work earnestly, and hope that the correction will meet with approval.

Once again, thank you very much for your comments and suggestions.

Yours

Sincerely

Qiang Li